# Effects of NaHCO_3_ Acclimation on Rye (*Secale Cereale*) Growth Under Sodic-Alkaline Stress

**DOI:** 10.3390/plants8090314

**Published:** 2019-08-30

**Authors:** Liyun Liu, Hirofumi Saneoka

**Affiliations:** Graduate School of Integrated Sciences for Life, Hiroshima University, 1-4-4 Kagamiyama, Higashi-Hiroshima 739-8528, Japan

**Keywords:** antioxidant capacity, ionic homeostasis, osmotic stress, root cell viability, sodium

## Abstract

Sodic-alkalinity is a serious limiting factor in agricultural productivity. This study was conducted to examine the contribution of acclimation to the adaptation of rye (*Secale cereale*) to sodic-alkalinity. Effects of acclimation were determined in two sets of experiments: One experiment for mineral accumulation, antioxidative capacity, and other physiological parameters; and a vivo experiment for root Evan’s Blue and Na^+^ influx from medium to root. Being exposed to sodic-alkalinity, acclimation did not affect plant dry weight. However, acclimation significantly reduced Na^+^ concentration and maintained a lower Na^+^/K^+^ ratio in all the tissues, increased the Ca^2+^ and Mg^2+^ concentrations in the root tissues, and increased the water uptake ability in comparison to the non-acclimated plants. Acclimation increased the antioxidant capacity represented by the increased activities of the enzymes SOD, GR, CAT, and GPOX in the leaf tissues of acclimated plants in comparison to the non-acclimated plants. Moreover, acclimation increased the root cell viability inhibited the Na^+^ influx to the root tissues in comparison to the non-acclimated plants. Together, these results suggest that rye can acclimate to sodic-alkalinity by increasing root cell viability, and therefore limited Na^+^ influx to root tissues and increased water uptake and antioxidant capacities without any change in the plant growth.

## 1. Introduction

The progressive increase of world population requires the agricultural productivity to keep up with the food demand. It is necessary to study the possibility of enhancing the crop productivity by improving the crop ability to cope with various environmental stresses. Acclimation is the phenomenon of plants improving their physiological ability to adapt to various environmental changes. Some plant species such as soybean, maize, pea, and mungbean, increased the ability to tolerate salinity after being exposed to low levels of stress for a period of time [1,2,3,4,5]. The better performances of acclimated plants are consequences of reduced accumulations of Na^+^ in soybean leaf tissues and in maize root tissues [1,5], better regulation of xylem ion loading (decrease of xylem Na^+^/K^+^) in pea [4], and enhancement of the antioxidant defense system and osmolyte accumulation in mungbean [3] under salinity. Normally, salinity is considered a soil with excessive NaCl or a mixture of NaCl and NaSO_4_, and sodic-alkalinity is considered a soil with high concentrations of NaHCO_3_ and Na_2_CO_3_ [6]. Sodic-alkalinity is widespread around the world, and more injurious to plant growth than salinity. Previous studies demonstrated the plant sodic-alkaline resistance was enhanced by foliar-sprayed 5-aminolevulinic acid, which also enhanced plant salt resistance [7,8]. Possibly, the same means as enhancing plant salt resistance can be applied to enhance sodic-alkaline resistance. Therefore, this study was conducted to investigate whether a low level of sodic-alkaline stress would enhance the sodic-alkaline tolerance of plants.

Salinity affects plant growth in two major stresses, osmotic and ionic stresses. The osmotic stress is an immediate phase that inhibits water uptake, cell expansion, and later bud development. The ionic stress develops lately when toxic ion Na^+^ is accumulated in plant tissues over the threshold [9]. Due to these two stresses, oxidative stress is often engendered under salinity [10]. More seriously, under sodic-alkalinity, the plentiful attendance of NaHCO_3_ and Na_2_CO_3_ results in a high soil pH (above 8.5), a high percentage of exchangeable sodium (>15%), a low water infiltration capacity, and a high bicarbonate level [11]. The high bicarbonate level surrounding the roots directly deteriorates the availability of several micronutrients, especially Fe, and it is often considered to be the primary factor for the chlorosis of plants under sodic-alkaline conditions [12]. Therefore, sodic-alkalinity exerts similar reactions but with the added effect of high pH caused by NaHCO_3_ or Na_2_CO_3_, is more detrimental to plant growth than salinity. To survive in sodic-alkaline conditions, plants have evolved a number of defense strategies, such as the accumulation of osmolytes to maintain low intracellular osmotic potential of plants, the maintenance of low cytoplasmic Na^+^ concentration in plant cells by exporting Na^+^ out of the cells and sequestrating Na^+^ within vacuoles, and the maintenance of cell membrane integrity by scavenging over-produced reactive oxygen species (ROS), to deal with sodic-alkalinity [13,14].

Na^+^ toxicity is one of the important factors inhibiting plant growth. The increased assimilation of Na^+^ interferes with K^+^, Ca^2+^, and Mg^2+^, resulting in nutrient deficiencies and ion imbalance in the tissues of plants [15]. Therefore, inhibiting Na^+^ accumulation in the plants, especially in the leaf tissues, is perhaps one of the most important Na^+^ stress resistance mechanisms in some halophytes and tolerant glycophytes to cope with salinity and sodic-alkalinity [9]. To achieve this, the exclusion of Na^+^ from the shoot by retrieving xylem Na^+^ into root xylem parenchyma cells is crucial. This excluded process significantly reduces the amount of Na^+^ transported in the xylem to the shoot and loads most plant Na^+^ at the root [16], which then helps plants maintain minimal cytosolic Na^+^/K^+^ ratios in the shoot. This exclusion ability is mediated by a number of plasma membrane-localized Na^+^/K^+^ transporters and Na^+^/H^+^ exchangers [17,18,19,20]. The efflux of Na^+^ from root cells is mediated by Salt-Overly-Sensitive-1 (SOS1) [21] such as *Arabidopsis* AtSOS1 [22]. Moreover, plant root surviving in the soil takes up water and minerals for their growth. However, high pH conditions impose severe root cell injury and death, which, consequently, limits mineral uptake [23]. Therefore, healthy root can maintain mineral uptake and protects plants.

Oxidative stress often occurs as a result of over-production of ROS, which damage various plant cells resulting in their dysfunction [24]. The detoxification of over-produced ROS is considered one of the important sodic-alkaline tolerance mechanisms. To detoxify ROS for alleviating oxidative stress damage, plants often employ antioxidants, including enzymatic and non-enzymatic mechanisms. Enhancement of antioxidant mechanism is associated with enhancement of sodic-alkaline tolerance [8]. Superoxide dismutase (SOD), one of the enzymatic antioxidants, catalyzes the dismutation of O_2_^−^ to H_2_O_2_, which is further detoxified to H_2_O and O_2_ by catalase (CAT), guaiacol peroxidase (GPOX), and ascorbate peroxidase (APX) [24,25]. Glutathione, ascorbate, carotenoids, phenolic compounds, and non-protein amino acids are considered the common non-enzymatic antioxidants [25,26].

Rye (*Secale cereale*) is a superior cereal grain to wheat or barley in healthy benefits. Rye contains various health benefits, including weight loss, prevention of gallstones and asthma, control of diabetes, improvement of digestion, boost in metabolic performance, and lowering of blood pressure [27,28,29,30]. Rye is classified in the salinity tolerance category [31]. However, the plant growth of rye was inhibited by bicarbonate and high pH, and this inhibition was not relieved by increasing the level of Zn supplementation [32]. Here, the study was conducted to investigate whether acclimation has the potential for improving rye sodic-alkaline tolerance through determining the osmotic stress, specific ion toxicity, and antioxidant activities.

## 2. Results

### 2.1. Plant Growth and Relative Water Content

At the end of the treatment, none of the acclimation treatments showed any differences to the control plants in leaf dry weight and sheath dry weight but showed a 35% reduction in root dry weight and 10% reduction in leaf relative water content (RWC) compared with the control plants (Figure 1). Acclimation did not increase the dry weights of acclimated plants, but increased the RWC by 8.5 and 9%, respectively, in the A-1 and A-2 conditions in comparison to the non-acclimation conditions.

### 2.2. Chlorophyll (Chl) a and b, and Total Phenolics Contents

The Chl a and b concentrations in the leaf tissues were the same in all the treatments (Figure 2). The total phenolic content in the leaf tissues was 22% higher in the non-acclimated plants, 21% higher in the 1 mM NaHCO_3_ acclimated plants, and 5% higher in the 2 mM NaHCO_3_ acclimated plants compared with the control plants. Thus, the total phenolic content in the leaf tissues was 14% lower in 2 mM NaHCO_3_ acclimated plants compared to non-acclimated plants.

### 2.3. Na^+^ Retention Ability Under Sodic-Alkaline Conditions

The Na^+^ concentration increased in all the different tissues in the non-acclimated plants (Figure 3A). The increase in Na^+^ was most pronounced in the root tissues, where the Na^+^ concentration was at least three times that in the leaf tissues and sheath tissues in the non-acclimated plants. Acclimation treatment decreased the Na^+^ concentrations in all the tissues of the plants. The Na^+^ concentration was 35, 38, and 20% lower in the leaf, sheath, and root tissues, respectively, in the 1 mM NaHCO_3_ acclimated plants compared to the non-acclimated plants. The Na^+^ concentration was 41, 26, and 15% lower in the leaf, sheath, and root tissues, respectively, in the 2 mM NaHCO_3_ acclimated plants compared to the non-acclimated plants. Thus, the plant Na^+^ accumulations were 34 and 30% lower, respectively, in the 1 mM NaHCO_3_ acclimated plants and 2 mM NaHCO_3_ acclimated plants compared to the non-acclimated plants (Figure 4A). 

The K^+^ concentration was 28, 18, and 82% lower in the leaf, sheath, and root tissues, respectively, in the non-acclimated plants compared to the control plants (Figure 3B). Acclimation treatment increased the K^+^ concentration by 20% in the leaf tissues of the 1 mM NaHCO_3_ acclimated plants compared to the non-acclimated plants. Acclimation treatment did not alter the plant K^+^ accumulation in either the 1 mM or 2 mM NaHCO_3_ acclimated conditions compared to the non-acclimated condition (Figure 4B). 

The significant increase in Na^+^ concentration and the significant decrease in K^+^ concentration, causing an increase in the Na^+^/K^+^ ratio (26-fold in the leaf tissues, 25-fold in the sheath tissues, and 117-fold in the root tissues), resulted in serious deterioration of the ionic homeostasis in the non-acclimated condition compared to the control plants (Figure 3C). The Na^+^/K^+^ ratio was 47 and 23% lower in the leaf tissues and root tissues in the 1 mM NaHCO_3_ acclimated plants compared to the non-acclimated plants, and 46 and 27% lower in the leaf tissues and root tissues in the 2 mM NaHCO_3_ acclimated plants compared to the non-acclimated plants, respectively. Acclimation significantly decreased the Na^+^/K^+^ ratio in the leaf and root tissues without any difference between the acclimated conditions, showing that acclimation prevented an ionic imbalance between Na^+^ and K^+^.

### 2.4. Root Ca^2+^ and Mg^2+^ Concentrations Under Sodic-Alkaline Conditions

The Ca^2+^ and Mg^2+^ concentrations in the leaf tissues were not different among all the plant growth conditions (Table 1). The Ca^2+^ concentration in the root tissues was not altered in the non-acclimated plants compared to the control plants, and 25 and 14% higher in the 1 mM NaHCO_3_ acclimated plants and 2 mM NaHCO_3_ acclimated plants compared to the non-acclimated plants. The Mg^2+^ concentration in the root tissues was decreased by 22% in the non-acclimated plants compared with the control plants, but it was completely restored in the 1 mM NaHCO_3_ acclimated plants and the 2 mM NaHCO_3_ acclimated plants.

### 2.5. Root Cell Viability and Root Na^+^ Efflux

The strongest Evan’s Blue staining of the root tips was observed in the non-acclimated plants compared to that in the acclimated plants after 10 mM NaHCO_3_ treatment for 30 min (Figure 5C). The highest quantification of Evan’s Blue extraction from the stained root tips was obtained in the control plants, and the quantification was reduced by 37.6, 68, and 68% in the non-acclimated, A-1, and A-2 plants compared that in control plants after 10 mM NaHCO_3_ treatment for 2 h (Figure 5B). These results mentioned that the acclimation can increase the root cell viability under sodic-alkaline conditions.

Root Na^+^ efflux ability was evaluated by measuring accumulated Na^+^ from medium by 10 mM NaHCO_3_-treated root (Figure 5A). The Na^+^ was accumulated mostly by non-acclimated and 0.2 mM NaHCO_3_-acclimated plants after 30 min 10 mM NaHCO_3_ treatment. However, the Na^+^ was most accumulated by non-acclimated, followed by 0.2 mM NaHCO_3_ acclimated plants and 0.1 mM NaHCO_3_ acclimated plants. 

### 2.6. Leaf Antioxidant Enzyme Activities

At the end of the treatments, the malondialdehyde (MDA) concentration was increased by 22.5% in the non-acclimated plants compared to the control plant, decreased by 27.1 and 29.4% in the 1 mM NaHCO_3_ acclimated plants and 2 mM NaHCO_3_ acclimated plants compared to the non-acclimated plants (Figure 6). The H_2_O_2_ concentrations was similar in all the plant growth conditions. The activities of SOD, CAT, GPOX, and glutathione reductase (GR) were similar between the control plants and the non-acclimated plants in the non-acclimated plants compared with the control plants (Table 2). Acclimation increased the SOD activity by 76% in the 1 mM NaHCO_3_ acclimated plants compared with the non-acclimated plants. The activities of CAT, GPOX, and GR were increased by 1.2-, 1-, and 1.8-fold in the 1 mM NaHCO_3_ acclimated plants compared with the non-acclimated plants. The activities of CAT, GPOX, and GR were increased by 0.9-, 0.9-, and 1.9-fold in the 2 mM NaHCO_3_ acclimated plants compared with the non-acclimated plants. Acclimation significantly increased the activities of SOD, CAT, GPOX, and GR in the leaf tissues without any difference between the acclimated conditions, showing that acclimation improved the antioxidative capacity of rye under a sodic-alkaline condition.

## 3. Discussion

Sodic-alkalinity leads to more severe damage to plants than salinity. Recently, some studies reported that a previous exposure to low-level salinity (acclimation) activates an array of processes leading to salinity tolerance improvements in different plant species such as soybean, rice, and pea [1,4,5,33]. However, the effects of acclimation on the sodic-alkaline of rye remain unknow. This study was conducted to demonstrate the possibility of rye sodic-alkaline tolerance induced by period NaHCO_3_ acclimation. Results showed that there was no effect of acclimation on the plant dry weight (Figure 1A). 

However, acclimation activates a set of physiological adjustments enabling the plant to survive in high sodic-alkaline conditions. In detail, acclimation increased the root cell viability (Figure 5A,B), inhibited the Na^+^ influx to root tissues (Figure 5C), decreased the toxic ion Na+ concentrations and Na^+^ accumulation in the plants (Figure 3A, Figure 4A), decreased the Na^+^/K^+^ ratio in all the plant tissues (Figure 3C), improved the water status in the leaf tissues (Figure 1B), and enhanced some antioxidant enzyme activities (Table 2) in the leaf tissues compared with the non-acclimated plants.

### 3.1. Detrimental Effects in Non-Acclimated Rye Under Sodic-Alkaline Conditions

Osmotic stress and ionic stress are the main toxic effects in the response of plants to an external Na^+^ environment. In the non-acclimated plants, the reduced RWC in the leaf tissues demonstrates the occurrence of the osmotic stress after the Na^+^ concentration in rhizosphere increased to a threshold level (Figure 1B), the abundant uptake of toxic ion Na^+^ in the root tissues demonstrates the occurrence of ionic stress resulting in the significant reduction in the root growth of rye (Figure 1A, Figure 3A). However, the occurrences of osmotic and ionic stresses did not inhibit the leaf growth of rye in the non-acclimated plants, even the Na^+^ concentration in the leaf tissues of rye was 20-fold higher compared with the control plants (Figure 1, Figure 3A). Possibly, rye has some mechanisms to restrict the detrimental effects of Na^+^ toxicity in the leaf tissues. The excessive cytoplasmic Na^+^ concentration may be detoxified via sequestration of Na^+^ within the vacuole, and therefore act as the major component of the cellular osmotic potential [14,15] in the leaf tissues of non-acclimated ryes. Furthermore, the abundant retention of Na^+^ in the root tissues of rye inhibited the transport of Na^+^ from root tissues to the sheath tissues and, finally, to the leaf tissues, and therefore offers another protection to the leaf tissues against Na^+^ toxicity in the non-acclimated plants (Figure 3A). Therefore, the maintenance of the shoot growth with marked increased Na^+^ concentration in the leaf tissues and the reduction of root growth due to the abundant Na^+^ retention in the root tissues suggest that ionic stress mainly occurred in the root tissue in the non-acclimated ryes. The ionic tolerance of rye might be improved via the reduction of Na^+^ uptake in the root tissues. 

Naturally, some main minerals such as K^+^, Ca^2+^, and Mg^2+^, assimilated by plants roots from the rhizosphere, are translocated upward to shoot through xylem. Under sodic-alkalinity, in this study, the increased Na^+^ concentration interfered with the acquisitions of K^+^ in all the plant tissues, especially in the root tissues (Figure 3B), and led to detrimental effects on the ion homeostasis between K^+^ and Na^+^ in the non-acclimated plants (Figure 3C). The excessive Na^+^ concentration and serious K^+^ deficiency in plant cells often causes overproduction of ROS, including H_2_O_2_ [9]. These ROS are associated with cell membrane damage and the product of membrane lipid peroxidation (MDA). The increases in MDA and H_2_O_2_ are clear signs of oxidative stress damage. In this study, the H_2_O_2_ concentration was similar in all the treatments. The maintains of the H_2_O_2_ concentration in the non-acclimated plants compared to the control plants, suggesting the absence of H_2_O_2_ in oxidative damage in the non-acclimated plants. However, the MDA concentration increased in the non-acclimated plants compared to the control plants. This increased level of MDA in the non-acclimated plants compared to those in the control plants (Figure 6) indicate the attendance of oxidative stress in the non-acclimated ryes. The maintenance of SOD, GR, CAT, and GPOX activities corresponded to the maintained H_2_O_2_ concentration in the non-acclimated plants (Table 2), suggesting the worthless of enzymatic antioxidants on the scavenging ROS in the non-acclimated ryes. The improved total phenolic compounds (non-enzymatic antioxidants) in the leaf tissues may contributed to the sufficient antioxidant capacity in the non-acclimated ryes (Figure 2B). 

### 3.2. Effect of Acclimation on the Na^+^ and K^+^ Concentrations in the Differential Tissues of Ryes

Acclimation has been used to several agricultural crops for the purpose of salt tolerance enhancement, including increased osmotic potential, improved antioxidant enzyme activity, improved K^+^ accumulation, improved vacuolar Na^+^ sequestration, and decreased Na^+^ accumulation [4,5,34]. In ryes, the RWC in the leaf tissues was increased, and the Na^+^ concentrations in all the tissues were decreased in the acclimated plants compared to that in the non-acclimated plants under sodic-alkaline stress (Figure 1B, Figure 3A). The increase of RWC in the leaf tissues and the decrease of Na^+^ concentrations in all the tissues of the acclimated plants demonstrates the stimulation of water uptake ability and the detoxification of Na^+^ toxicity compared to that in the non-acclimated plants. The decrease of Na^+^ concentration also contributed to the balance between Na^+^ and K^+^ and therefore maintained better cellular Na^+^/K^+^ homeostasis for plant survival in the acclimated plants than in the non-acclimated plants (Figure 3C). Normally, ion homeostasis during sodic-alkaline stress also requires the maintenances of stable K^+^ acquisition and distribution to balance the toxic effects of Na^+^ accumulation [14]. In ryes, acclimation did not improve the K^+^ concentration in all the plant tissues compared to that in the non-acclimated plants (Figure 3B). Hence, the beneficial effect of acclimation in rye is achieved by regulating the water uptake and Na^+^ accumulation mechanisms. 

For Na^+^ concentration, the acclimated plants showed a higher capacity to exclude sodium in the leaf and sheath tissues than the root tissues (Figure 3A). In the acclimated plants, the Na^+^ accumulation was decreased more in the shoot tissues (>30%) compared to the root tissues (<20%) (Figure 4A). Munns and Tester [9] suggests a lower accumulation of Na^+^ in the shoot tissues is important for cellular metabolism under a high external Na^+^ environment and, therefore, the exclusion of Na^+^ from the shoots is the most essential feature of high Na^+^ tolerance in plants. To exclude Na^+^ from the shoot, a plant can either inhibit Na^+^ entry from the root or retrieve xylem Na^+^ into root xylem parenchyma cells. This Na^+^ exclusion process is mediated by a number of plasma membrane-localized Na^+^/K^+^ transporters and Na^+^/H^+^ exchangers [17,18,19,20]. In the acclimated plants, the Na^+^ concentration was decreased more in the leaf and sheath tissues compared to the root tissues. This more decreased Na^+^ concentration in the leaf and sheath tissues demonstrates that acclimation may improve more mechanisms for Na^+^ exclusion from the leaf tissues to the root tissues in the acclimated plants compared to the non-acclimated plants. This Na^+^ exclusion process from the shoot reduces the amount of Na^+^ transported in the xylem to the shoot and loads most plant Na^+^ in the root tissues, which helps ryes maintain minimal cytosolic Na^+^/K^+^ ratios in the leaf tissues. 

Amount Na^+^ loading in the root tissues still causes various detrimental effects on the root growth. In the acclimated plants, although the root growth was non altered, the Na^+^ concentration in the root tissues was decreased by 20% compared to that in non-acclimated plants (Figure 1A, Figure 3A). Therefore, the acclimation to sodic-alkalinity is attributed to the better ability of rye roots to exclude Na^+^ to rhizosphere, given the significant difference in Na^+^ fluxes between the acclimated roots and the non-acclimated roots (Figure 3 and Figure 4). In *Arabidopsis*, Na^+^ exclusion from the root cytosol to the external medium under a high external Na^+^ environment is mediated by the operation of the plasma membrane Na^+^/H^+^ exchangers encoded by SOS1, and the overexpression of SOS1 contributes to inhibit Na^+^ accumulation and improve the salinity tolerance [22,35]. Possibly, acclimation may also trigger other Na^+^ transport mechanisms for Na^+^ exclusion from the root tissues to the external medium. This Na^+^ exclusion process from the root reduces the amount of Na^+^ transported to the root, which helps ryes maintain minimal cytosolic Na^+^/K^+^ ratios in the root tissues of acclimated plants. Moreover, in this study, there is a more inhibition of root Na^+^ influx ability in A-1 acclimated plants than A-2 acclimated plants compared with in non-acclimated plants, and this inhibition effectively controlled the Na^+^ assimilation by the root from medium (Figure 5C). This inhibition ability may because of the enhancement of root cell viability, which was appeared as color-fading of the Evan’s Blue staining and Evan’s Blue optical density of the root tips (Figure 5A,B). 

Taken together, 1 and 2 mM NaHCO_3_ stimulated water uptake in the leaf tissues and the ionic balance between Na^+^ and K^+^ in all the tissues, indicating the possibility of improving rye sodic-alkaline tolerance. Na^+^ exclusion from uptake is considered to be the most crucial trait contributing to sodic-alkaline tolerance in the acclimated rye. 

### 3.3. Effect of Acclimation on the Ca^2+^ and Mg^2+^ Concentrations in the Root Tissues of Ryes

In the acclimated plants, the reduced transport of Na^+^ to the root tissues did not increase the uptake of K^+^ in all the plant tissues (Figure 3). However, it was found that the reduced transport of Na^+^ to the root tissues from soils increased the uptake of other elements, such Ca^2+^, whose concentration in the root tissues was increased, and Mg^2+^, whose concentration in the root tissues was restored compared with in the control plants (Table 1). The increased uptake of Ca^2+^ and Mg^2+^ maintained ion homeostasis in the root tissues, which may not be crucial for plant survival but is an essential process for plant growth in the acclimated plants. The increased Ca^2+^ concentration in the root tissues can contribute to the requirement of various structural roles in the cell wall and membranes, and partially prevent cell damage in the root tissues of acclimated plants. However, the increase of Ca^2+^ concentration in the root tissues did not alter the Ca^2+^ concentration in the leaf tissues of acclimated plants compared with in the non-acclimated plant (Table 1). The cell membrane damage in the leaf tissues was prevented by the improved antioxidant enzyme activities in the acclimated plants (Table 2; Figure 6). Cellular Mg^2+^ is a mobile nutrient which though passes the phloem for recycling in plants and is the central atom of chlorophyll molecules in green tissues [36]. The restoration of the Mg^2+^ concentration in the root tissues of acclimated plants may enable rye to resume chlorophyll biosynthesis in the leaf tissues to the same level as that in the control plants. 

### 3.4. Effect of Acclimation on the Antioxidant Capacity in the Leaf Tissues of Ryes

The main consequences of plant exposure to excessive external Na^+^ are water deficit and Na^+^ excess, which cause osmotic stress and ionic stress. The osmotic stress and ionic stress strongly inhibit the assimilation of photosynthetic carbon and then cause an over-reduction of photosynthetic electron transport activity [25]. The reduction of electron transport activity causes the over-production of ROS, which lead to oxidative damage in plants [10]. In this study, it was found that the MDA concentration was decreased in the acclimated plants compared to those in the non-acclimated plants (Figure 6), which suggests that acclimation suppress the oxidative stress in the leaf tissues of acclimated plants compared to the non-acclimated plants. The rise in water uptake (Figure 1) and the reduction in Na^+^ concentration (Figure 3) in the leaf tissues in the acclimated plants demonstrates the reductions in osmotic stress and ionic stress compared to that in the non-acclimated plants. Those reductions in osmotic stress and ionic stress may suppress the overproduction of ROS in the acclimated plants compared to that in the non-acclimated plants. 

Protection of plants against stress-induced ROS is also achieved by means of scavenging the ROS already produced [37]. To scavenge ROS, plants often employ the non-enzymatic antioxidants, such as chlorophyll and phenolic compounds, and the enzymatic antioxidants, such as SOD, GR, CAT, and GPOX [25,26,38,39]. The enhanced synthesis of chlorophyll and total phenolic compounds and the increase in enzyme activities under environmental stress are often correlated with reduced oxidative damage and, therefore, increased stress tolerance [40,41]. In ryes, the non-alterations of chlorophyll and total phenolic concentrations in the leaf tissues of acclimated plants (Figure 2) demonstrates the worthless of their functions on ROS scavenging activity compared to that non-acclimated plants. However, the activities of SOD, GR, CAT, and GPOX markedly increased in the leaf tissues of acclimated plants (Table 2), which is responsible for suppressing the oxidative damage in the leaf tissues of acclimated plants (Figure 6). The enzyme SOD is considered to be the vital first line of defense against ROS [26]. This enzyme is responsible for scavenging O_2_^−^ to H_2_O_2_, which are further scavenged to H_2_O and O_2_ by GR, CAT, APX, and GPOX [27,28]. The significantly increased SOD activity participated in strongly adjusting the low O_2_^−^ state in the acclimated plants. The enzyme GR is localized predominantly in the chloroplast, where plants conduct photosynthesis [27]. The increase in GR activity strongly protected the chloroplast against oxidative toxicity, and consequently minimized lipid peroxidation in the acclimated plants compared with that in the non-acclimated plants (Table 2). Moreover, the enzyme CAT directly scavenges H_2_O_2_ into H_2_O and O_2_, and the enzyme GPOX contributes to defense against oxidative stress by scavenging H_2_O_2_ [27,28]. In this study, the increased MDA concentration in the non-acclimated plants compared to those in the control plants suggested the oxidative damage in the non-acclimated ryes. However, the H_2_O_2_ concentration was the same between the non-acclimated plants and acclimated plants. Therefore, the oxidative damage should be caused by another ROS, like O_2_^−^. In the acclimated plants, the increased SOD activity in the leaf tissues adjusted the O_2_^−^ to produce more H_2_O_2_, and further the significant increases in the activities of CAT and GPOX strongly contributed to the scavenging H_2_O_2_ and, therefore, improved the antioxidative ability in the acclimated ryes. 

An occurrence in oxidative damage may be a result of elevated ROS generation and a decrease in the scavenging capacity of antioxidants. In plants, partial suppression of ROS production and scavenging of the ROS already produced protect plants against stress-induced oxidative stress [37]. Possibly, the reduced osmotic stress and ionic stress suppressed the overproduction of ROS, and the increased antioxidative enzyme activities scavenged the over-produced ROS in the acclimated plants. Therefore, the slight reduction in oxidative stress is achieved not only by the reduced osmotic and ionic stresses, but also the increased antioxidative enzymes activities in the acclimated plants (Figure 1B, Figure 3A, Table 2).

## 4. Materials and Methods 

### 4.1. Plant Material and Treatment Conditions

The seeds of rye (*Secale cereale*), cultivar Haruichiban, were obtained from the Snow Brand Seed Corporation Limited, Japan. The seeds were sown in a seedbed in the greenhouse of the Faculty of Applied Biological Sciences, Hiroshima University, Japan (in February). At 20 days after germination, seedlings were transferred to 350-cm^3^ pots (two seedlings per pot) filled with healthy soil containing substrate and irrigated regularly with nutrient solution. The composition of this nutrient solution was 1.6 mM N, 0.68 mM K, 0.38 mM Ca, 0.2 mM P, 0.2 mM Mg, 8 µM B, 5 µM Fe, 4 µM Mn, 0.3 µM Zn, 0.116 µM Cu, and 0.07 µM Mo. Four-week-old seedlings were divided into four groups (four pots per group). Two groups were irrigated with either 1 mM NaHCO_3_ or 2 mM NaHCO_3_ (pH 7.1) (acclimation). Ten days later, all the pots were irrigated with 50 mM NaHCO_3_ for 5 days, and then 100 mM NaHCO_3_ for the other 5 days, except for one group, which remained as a control. The condition of the greenhouse was 55% humidity, 18–25 °C day/10–17 °C night temperature and natural sunlight. The four treatments are abbreviated as follow:

Control---no acclimation and no treatment

NA--------no acclimation, 100 mM NaHCO_3_ treatment

A-1--------1 mM NaHCO_3_ acclimation, 100 mM NaHCO_3_ treatment

A-2--------2 mM NaHCO_3_ acclimation, 100 mM NaHCO_3_ treatment

The pH of irrigated nutrient medium was 6.10 for control. The pH range of irrigated nutrient medium was 7.43 and 7.67 after adding 1 and 2 mM NaHCO_3_ for acclimation, and 8.0 and 8.17 after adding 50 and 100 mM NaHCO_3_ for true treatment, respectively.

### 4.2. Plant Growth and Relative Water Content Measurements

After 10 days of true sodic-alkalinity treatment, the plants were harvested at a position just above the cotyledon following the separation of leaves, sheaths, and roots, which were oven-dried for recording dry weight. The young leaf, sheath, and root tissues were flash-frozen in liquid nitrogen and stored at −80 °C until use for analyses. The leaf relative water content was calculated using the equation 100 × (fresh weight-dry weight)/(turgor weight-dry weight). 

### 4.3. Chlorophyll and Total Phenolics Concentrations

Chl was estimated from fresh young leaf tissues with N,N-dimethylformamide, and the absorbance of the extract was measured at 647 and 664 nm; next Chl *a* and *b* concentrations were calculated following the method described by Inskeep and Bloom [42]. For the total phenolics concentration, a methanolic extraction (after centrifugation at 10,000× *g* for 10 min) of fresh leaf tissues was mixed with 400 mM Na_2_CO_3_ and double-diluted Folin-Ciocalteau reagent, and the absorbance of the mixture was recorded at 765 nm after 5 min incubation at 50 °C [43]. Gallic acid was used as a standard for calculation.

### 4.4. Mineral Analysis

Approximately 0.2 g freeze-dried young plant tissues were ground to fine powder and digested with HNO_3_-H_2_O_2_ for the measurements of Na^+^ and K^+^ (using a flame photometer: ANA-135, Eiko Instruments Inc., Tokyo, Japan) in the leaf, sheath, and root tissues, and Ca^2+^ and Mg^2+^ (using an atomic absorption spectrophotometer: U-3310 Hitachi Co. Ltd., Tokyo, Japan) in the leaf and root tissues. 

### 4.5. Oxidative Stress Measurements

For H_2_O_2_ concentration, a methanolic extraction of leaf tissues was mixed with reaction buffer (25 mM H_2_SO_4_, 0.25 mM FeSO_4_, 0.25 mM (NH_4_)_2_SO_4_, 10 mM sorbitol, and 125 μM xylenol orange), and absorbance of the mixture was recorded at 560 nm after 1 h incubation at 25 °C [44]. The H_2_O_2_ concentration was used as a standard for calculation. The MDA concentration in young leaf tissues was determined using a modified version of the thiobarbituric acid procedure [45], and the MDA concentration was calculated using 1.55 mM^−1^ cm^−1^ as extinction coefficient.

### 4.6. Antioxidant Enzyme Assays

The extraction method of soluble antioxidant enzymes from the leaf blade tissues was the same as that described by Liu et al. [7]. For measuring total soluble SOD activity, one set of reaction mixtures, which contained 50 mM potassium phosphate buffer (pH 7.8), 0.1 mM ethylenediaminetetraacetic acid, 13 mM methionine, 75 mM nitroblue tetrazolium, 2 mM riboflavin, and 5% enzyme extract, was illuminated with 25 W fluorescent lamps for 15 min, and the other set was kept in the dark as a control. The absorbance was measured at 560 nm after reaction, and one unit of SOD activity was defined as the amount of enzyme required to cause 50% inhibition of nitroblue tetrazolium photoreduction [46]. For measuring total GR activity, the GR assay mixture contained 40 mM potassium phosphate buffer (pH 7.5), 0.78 mM glutathione, 0.4 mM ethylenediaminetetraacetic acid, 0.02 mM nicotinamide adenine dinucleotide phosphate, and 10% enzyme extract. The NADPH oxidation was monitored at 340 nm, and GR activity was calculated using the molar extinction coefficient for NADPH (6.22 mM^−1^ cm^−1^) [47]. For measuring total CAT activity, the CAT assay mixture contained 10 mM H_2_O_2_, 5% enzyme extract, 50 mM potassium phosphate buffer (pH 7.0), and CAT activity was calculated using the molar extinction coefficient for H_2_O_2_ (0.04 mM^−1^ cm^−1^) [48]. For total soluble GPOX activity, the GPOX assay mixture contained 70 mM potassium phosphate buffer (pH 7.0), 15 mM guaiacol, 10 mM H_2_O_2_, and 5% enzyme extract. GPOX activity was calculated using the molar extinction coefficient for tetra guaiacol (26.6 mM^−1^ cm^−1^) [49]. The protein concentration in the enzyme extract was measured using a Protein Assay kit (Bio-Rad DC, CA, USA) and bovine serum albumin as standard, according to the manufacturer’s instructions.

### 4.7. Growth for Root Na^+^ Efflux and Root Damage in Hydroponic Condition

The seeds of rye were surfaced-sterilized for 15 min immersion in 1% sodium hypochlorite and then completely rinsed with deionized water. Seeds were germinated on the plastic net floating on 20 L of tap water, and nutrient solution after germination. The composition of this nutrient solution was the same as the greenhouse experiment. At 10 days of germination, uniform seedlings were selected and transferred to 1.3 L plastic containers containing the same nutrient solution. The root tissues can be injured more in hydroponic condition at same saline level compared to soil condition, and the age of rye seedling was younger in hydroponic condition compared to soil condition. Therefore, ten times lower NaHCO_3_ was provided in pre-treatment and treatment in hydroponic condition compared to soil condition. Twenty-four seedlings were placed in each container, two containers were pre-treated with either 0.1 mM NaHCO_3_ or 0.2 mM NaHCO_3_ (pH 7.1–7.2) (acclimation) and the other two containers were still grown in nutrient solution. Five days later, one container grown in the nutrient medium was kept as control and the remaining containers were treated with 10 mM NaHCO_3_ (pH 8.05).

### 4.8. Root Na^+^ Efflux

Na^+^ efflux was determined as the description by Hamam et al. [20] with minor modification. The root tissues from non-acclimated and acclimated plants were kept in 5 cm^3^ microtubes with 10 mM NaHCO_3_ for 30 min and 3 h, the root tissues from control were still kept in nutrient medium, and then Na^+^ concentration in the medium was determined by a flame photometer (ANA-135, Eiko Instruments Inc., Tokyo, Japan). Accumulated Na^+^ by plants was calculated using the equation 10 mM-residual Na^+^ in the medium after 10 mM NaHCO_3_ treatment. 

### 4.9. Root Cell Viability

Root tips (3 cm) were infiltrated with a 0.25% (*w/v*) aqueous solution of Evan’s Blue for 30 min after 10 mM NaHCO_3_ treatment (30 min), and then were rinsed with deionized water until no further blue dye was eluted from the roots [22]. The stained root tips were observed by an Axio Imager Z1 microscope Zeiss (Jena, Germany). For quantification of the Evan’s Blue staining, the root tips pre-treated seedlings were infiltrated with a 0.25% aqueous solution for 1 h after 10 mM NaHCO_3_ treatment (2 h). The stained root tips were solubilized by incubating in 1% (*w/v*) SDS in 50% (*v/v*) methanol at 50 °C for 1 h, and absorbance was measured at 600 nm.

### 4.10. Statistical Analyses

All data collected were subjected to one-way analysis of variance (ANOVA) using the IBM SPSS statistical package version 25. Significance testing was performed using Duncan’s multiple range test at *p* ≤ 0.05. The values are means (±standard error) of five replicates. 

## 5. Conclusions

The results in this study suggest the possibility of the acclimation to alleviate detrimental effects in ryes to sodic-alkalinity. The mitigating effects of acclimation in ryes could be attributed to the remarkable enhancement of the root cell viability, which is helpful for the inhibition of Na^+^ influx to the root tissues and Na^+^ detoxification in the plant tissues. Moreover, the increase in water uptake in the acclimated leaves prevents the cell dehydration, and the enhanced activities of certain ROS-scavenging enzymes sustains the antioxidant defense system, whereby ROS scavenging and maintenance of membrane integrity permitted the normal metabolic function.

## Figures and Tables

**Figure 1 plants-08-00314-f001:**
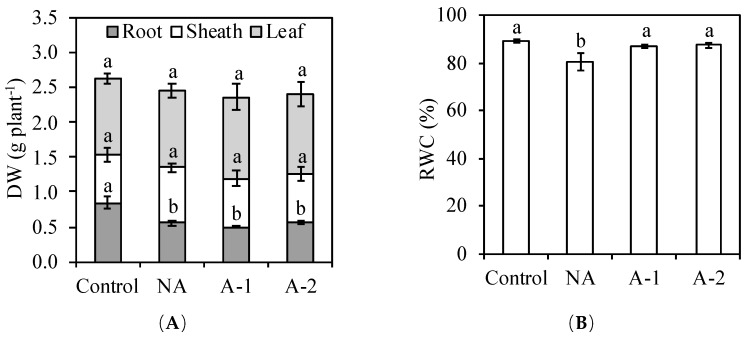
Effects of 10-days sodic-alkalinity with 100 mM NaHCO_3_ on plant dry weight (DW) (**A**) and leaf relative water content (RWC) (**B**) of pre-acclimated rye. Rye plants were pre-acclimated by 0 mM NaHCO_3_ (NA) and 1 and 2 mM NaHCO_3_ (A-1, A-2) for 10 days. Control plants were consistently irrigated by nutrient solution. Significance testing was performed using ANOVA analysis followed by Duncan’s multiple range test at *p* ≤ 0.05. The values are means (±SE) of five replicates. Means followed by the same letter are not significantly different.

**Figure 2 plants-08-00314-f002:**
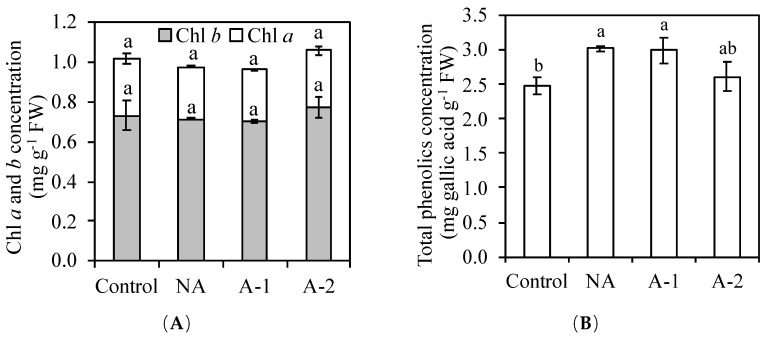
Effects of 10-days sodic-alkalinity with 100 mM NaHCO3 on chlorophyll (Chl) *a* and *b* contents (**A**) and total phenolics content (**B**) in the leaf tissues of pre-acclimated rye. Rye plants were pre-acclimated by 0 mM NaHCO_3_ (NA) and 1 and 2 mM NaHCO_3_ (A-1, A-2) for 10 days. Control plants were consistently irrigated by nutrient solution. Significance testing was performed using ANOVA analysis followed by Duncan’s multiple range test at *p* ≤ 0.05. The values are means (±SE) of five replicates. Means followed by the same letter are not significantly different.

**Figure 3 plants-08-00314-f003:**
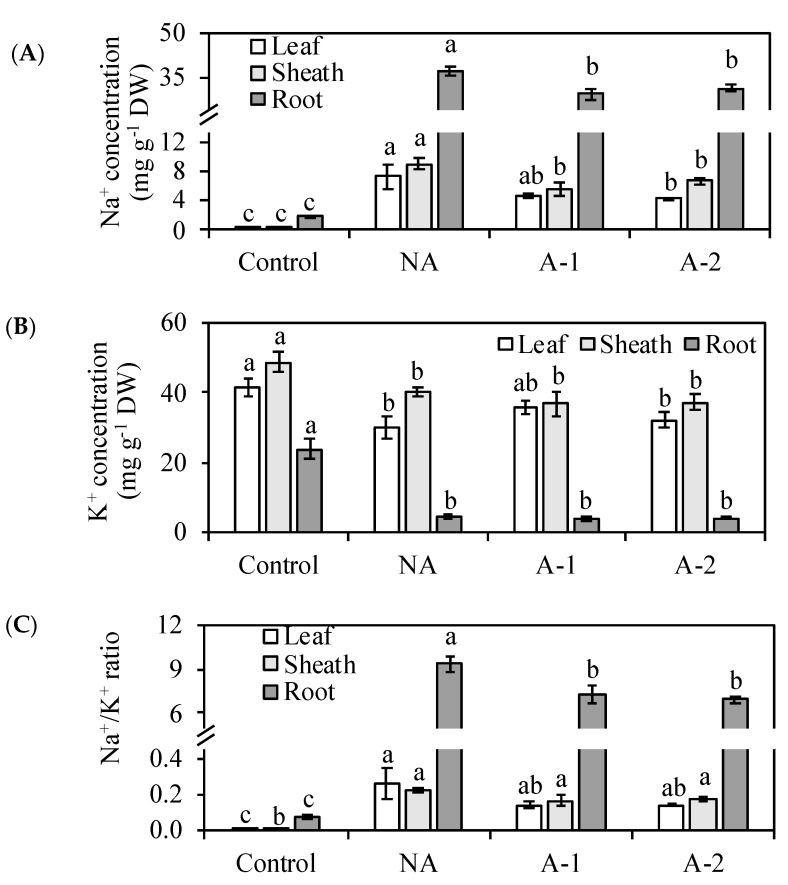
Effects of 10-days sodic-alkalinity with 100 mM NaHCO_3_ on Na^+^ concentration (**A**), K^+^ concentration (**B**), and Na^+^/K^+^ ratio (**C**) of pre-acclimated rye. Rye plants were pre-acclimated by 0 mM NaHCO_3_ (NA) and 1 and 2 mM NaHCO_3_ (A-1, A-2) for 10 days. Control plants were consistently irrigated by nutrient solution. Significance testing was performed using ANOVA analysis followed by Duncan’s multiple range test at *p* ≤ 0.05. The values are means (±SE) of five replicates. Means followed by the same letter are not significantly different.

**Figure 4 plants-08-00314-f004:**
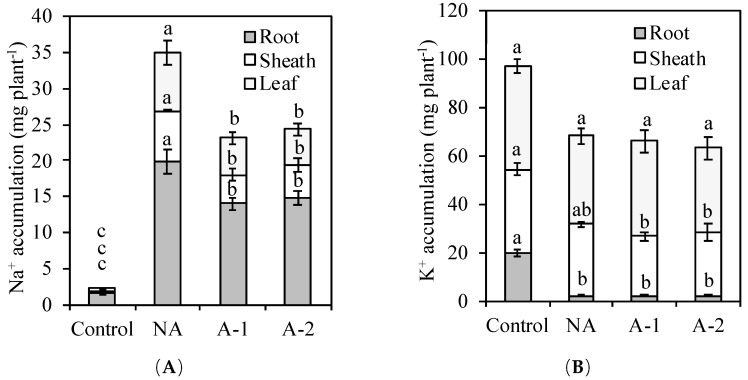
Effects of 10-days sodic-alkalinity with 100 mM NaHCO_3_ on Na^+^ accumulation (**A**) and K^+^ accumulation (**B**) of pre-acclimated rye. Rye plants were pre-acclimated by 0 mM NaHCO_3_ (NA) and 1 and 2 mM NaHCO_3_ (A-1, A-2) for 10 days. Control plants were consistently irrigated by nutrient solution. Significance testing was performed using ANOVA analysis followed by Duncan’s multiple range test at *p* ≤ 0.05. The values are means (±SE) of five replicates. Means followed by the same letter are not significantly different.

**Figure 5 plants-08-00314-f005:**
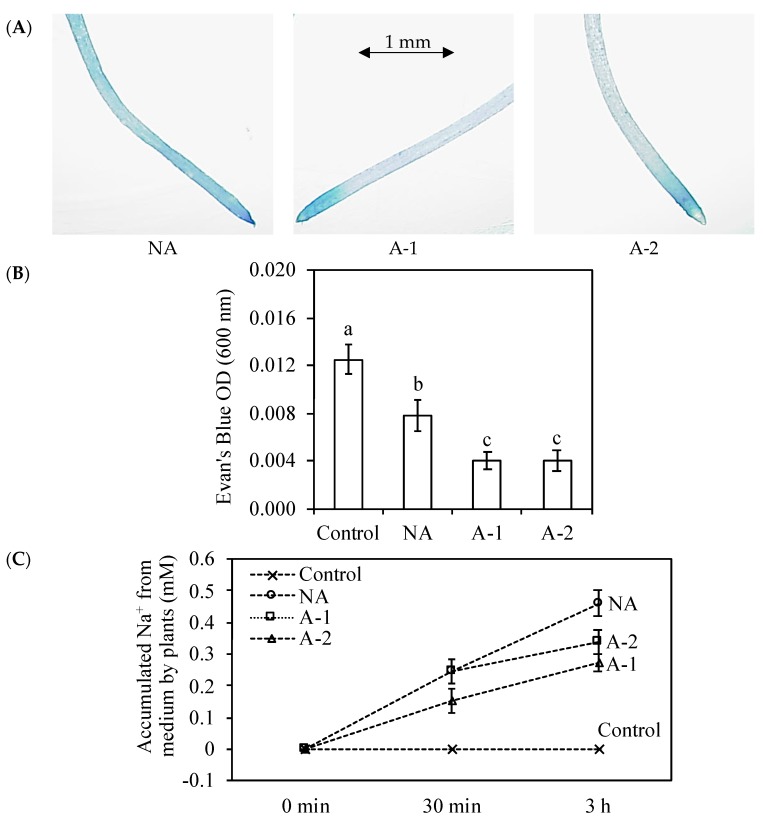
Effects of 30 min sodic-alkalinity with 10 mM NaHCO_3_ on the Evan’s Blue staining of the root tips of pre-acclimated rye (**A**), and effects of 2 h sodic-alkalinity with 10 mM NaHCO3 on the quantification of the Evan’s Blue of the root tips of pre-acclimated rye (**B**). Effects of 30 min and 3 h sodic-alkalinity with 10 mM NaHCO_3_ on the Na^+^ accumulation of pre-acclimated rye (**C**). Rye plants were pre-acclimated by 0 (NA) and 0.1 and 0.2 mM NaHCO_3_ (A-1, A-2) for 5 days. Control plants were consistently irrigated by nutrient solution. Significance testing was performed using ANOVA analysis followed by Duncan’s multiple range test at *p* ≤ 0.05. The values are means (±SE) of five replicates. Means followed by the same letter are not significantly different. OD means the optical density of samples measured at a 600 mm wavelength.

**Figure 6 plants-08-00314-f006:**
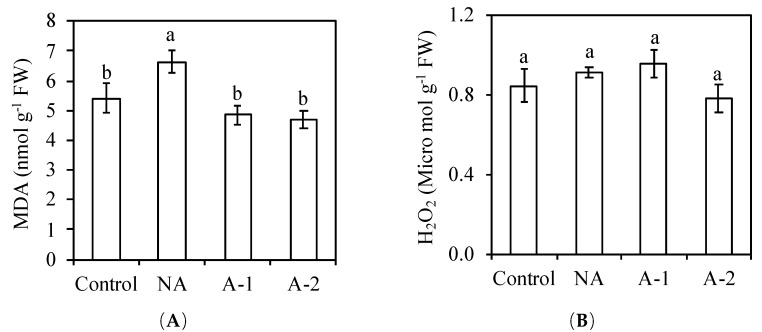
Effects of 10-days sodic-alkalinity with 100 mM NaHCO_3_ on malondialdehyde (MDA) (**A**) and hydrogen peroxide (H_2_O_2_) (**B**) of pre-acclimated rye. Rye plants were pre-acclimated by 0 mM NaHCO_3_ (NA) and 1 and 2 mM NaHCO_3_ (A-1, A-2) for 10 days. Control plants were consistently irrigated by nutrient solution. Significance testing was performed using ANOVA analysis followed by Duncan’s multiple range test at *p* ≤ 0.05. The values are means (±SE) of five replicates. Means followed by the same letter are not significantly different.

**Table 1 plants-08-00314-t001:** Effects of 10-days sodic-alkalinity with 100 mM NaHCO_3_ on Ca^2+^ and Mg^2+^ concentrations in the leaf and root tissues of pre-acclimated rye. Rye plants were pre-acclimated by 0 mM NaHCO_3_ (NA) and 1 and 2 mM NaHCO_3_ (A-1, A-2) for 10 days. Control plants were consistently irrigated by nutrient solution.

Measurement		Control	NA	A-1	A-2
Ca^2+^	Leaf	0.94 ± 0.13 ^a^	0.82 ± 0.08 ^a^	0.81 ± 0.03 ^a^	0.80 ± 0.09 ^a^
	Root	1.24 ± 0.03 ^b^	1.12 ± 0.03 ^b^	1.40 ± 0.07 ^a^	1.27 ± 0.06 ^ab^
Mg^2+^	Leaf	1.36 ± 0.11 ^a^	1.45 ± 0.17 ^a^	1.32 ± 0.09 ^a^	1.44 ± 0.21 ^a^
	Root	1.18 ± 0.13 ^a^	0.92 ± 0.02 ^b^	1.17 ± 0.05 ^a^	1.21 ± 0.04 ^a^

Significance testing was performed using ANOVA analysis followed by Duncan’s multiple range test at *p* ≤ 0.05. The values are means (±SE) of five replicates. Means followed by the same letter are not significantly different.

**Table 2 plants-08-00314-t002:** Effects of 10-days sodic-alkalinity with 100 mM NaHCO_3_ on superoxide dismutase (SOD), glutathione reductase (GR), catalase (CAT), and guaiacol peroxidase (GPOX) activities in the leaf tissues of pre-acclimated rye. Rye plants were pre-acclimated by 0 mM NaHCO_3_ (NA) and 1 and 2 mM NaHCO_3_ (A-1, A-2) for 10 days. Control plants were consistently irrigated by nutrient solution.

Treatments	SOD Activity (Unit mg^−1^ Protein)	GR Activity (mmol min^−1^ mg^−1^ Protein)	CAT Activity (mmol min^−1^ mg^−1^ Protein)	GPOX Activity (mmol min^−1^ mg^−1^ Protein)
Control	1114.63 ± 160.98 ^b^	27.75 ± 2.47 ^b^	6.48 ± 0.66 ^b^	87.68 ± 25.63 ^b^
NA	1148.80 ± 241.32 ^b^	16.89 ± 5.26 ^b^	6.21 ± 1.42 ^b^	86.62 ± 14.29 ^b^
A-1	2026.54 ± 138.03 ^a^	46.70 ± 3.38 ^a^	13.64 ± 0.40 ^a^	177.24 ± 25.01 ^a^
A-2	1254.31 ± 135.42 ^b^	49.63 ± 3.88 ^a^	11.92 ± 1.02 ^a^	165.01 ± 17.05 ^a^

Significance testing was performed using ANOVA analysis followed by Duncan’s multiple range test at *p* ≤ 0.05. The values are means (±SE) of five replicates. Means followed by the same letter are not significantly different.

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
