# Peer review of "Effects of NaHCO3 Acclimation on Rye (Secale Cereale) Growth Under Sodic-Alkaline Stress"

_plants, 2019, doi:10.3390/plants8090314_

Round 1

Reviewer 1 Report

General comments:

 The authors report on the observation that the acclimation of 1 and 2 mM NaHCO3for 10 days helps to Rye to adapt sodic alkalinity stress. Acclimation reduces Na+ concentration in the tissues and keeps a lower level of Na+/K+ ratio. Moreover, acclimation increased antioxidant capacity and root cell viability.

In general, the manuscript is well written and presented. However, it is unclear if the concentration of 100 mM NaHCO3 used in the experiments was enough to produce oxidative damage in the plants. The results showed that the accumulation of H2O2 and MDA were not different among all the treatments, means that is not clear oxidative damage in the plants.

Some comments are provided below regarding some suggestions about how the manuscript might be improved. My general recommendation for the authors is to improve the uniformity in the text and the figures. It is not clear if you are measuring concentration or content in some of your traits.

Specific comments

Keywords:

Osmotic stress should be in capital letters to be consistent with the other words.

Introduction:

I think is necessary for one or two sentences different what is salinity and sodic-alkalinity in the soil before to explain the effects plants in the line (41). Probably those sentences could be good in the line 35.

Line 31 to 35: It is not clear to which environment stress is referred to in this sentence (Sodic-alkalinity, salinity, drought, heat ??)

Line 49: After the word Therefore is, and not (.).

Line 67-68: The sentence “Plant root anchors plant roots to the soil ..” need to be redacted in a better way.

Line 72-73: Should be “.. is considered one of the most important…” or “is considered one of the important…” the first one sounds better.

Line 75: Needs to add a reference.

Line 80: Rye is a superior cereal grain to wheat or barley…. But in what? Is yield superior? Healthy benefits? Agronomy?

Line 80-82: Needs to add at least two references to show that all the health benefits

Line 84-95. Two sentences could merge, both ideas came from the same reference.

Results:

Figure 1: In the legend is confused when you mention that you pre-acclimated by 0 mM NaHCO3 in the control, you should leave just (NA), control is assumed that you did not put anything. The symbol = SE should be ± SE. The P-value in the material section you expressed such p ≤ 0.05 should be the same in the figures.

Same comments for all the figures and tables.

Figure 2A: In the y-axis, add content after Total phenolics

Line 102:  Be clear if you measure content or concentration?? For your case should be chlorophyll content.

Figure 3. The y-axis should be concentration and not content. In the text, you are mention Na and K concentration and not content. It is concentration or content?

Line 114: It should be Fig. 3A

Line 123: Change to Fig. 4A

Line 132: Change to Fig. 3B

Line 133: The concentration of K+ increased in relation to NA, right? You should merge this sentence with the following sentence in just one.

Line 137: Change to Fig. 4B

Line 141: Change to Fig. 3C

In the Figure 4 check the letters in both graphs (A-B). I think in the Fig. 4A, the treatments A-1 and A-2 for leaf tissue has to be the letter “b”, the same in the treatments NA, A-1, and A-2 for the leaf tissue in the Fig. 4B. Check again the stats to confirm.

Line 152: Change contents by concentrations. I am still confused if it is content or concentration?

Line 157: Why you put in parenthesis the percentage, previously you are reporting the increased and reduction without parenthesis, has to be consisted.

Figure 5. I recommend to change the order of the figures to flow the text, thus; the figure 5C should be 5A, and 5A change to 5C. Seems the legend not corresponded with the graphs, check it again. In figure 5B, what is OD? Explain in the legend

Line 165 and 168: Included the time of the treatments.

Line 182 to 184. Improve the redaction is confused (more/most), more is considered as a comparative form.

Line 187. MDA is the first time that shows in the text, you should indicate its mean.

Line 188. GR has to spell everything. It’s the first time in the text.

Lines 190 – 193: Why do you put in parenthesis?  If you start to present the increased or reduction without parenthesis.

Discussion:

Line 212 and 213: The figures mention in the text don’t correspond with the current figure 5.

Line 268: It has to mention under sodic-alkaline stress.

Line 280: Change: “The Na+ accumulation was more decreased” by The Na+ accumulation was decreased more

Line 281: It was decreased in comparison to what?

Line 287-290: Redact better. It is hard to understand this sentence.

Line 306: “that in non-acclimated” change by within non-acclimated

Line 307:  Add “the” before root.

Line 307 to 309: The figures mention in the text don’t correspond with the current figure 5.

Line 309: Again what is OD?

Line 319 and 235: Change “compared to that in” by compared within.

Line 337: H2O2 concentration was the same in acclimated plants not just with NA treatment but also with the control. Seems that the 100 mM NaHCO3 did not produce any oxidative stress. However, there is high enzymatic antioxidants activity just in the acclimate treatments, how could you explain that?

Line 350: It is concentration or content?

Line 362: Needs reference.

Line 363-365: What happened with NA, there is not increased of any antioxidant enzyme and the levels of H2O2 are the same as the control.

Materials and Methods:

Line 379: What do you mean with healthy soil? What kind of soil? Was it sterilized? Or it was a substrate?

Line 383: You should say “all the pots were irrigated” not the seedings

Line 375: You should include more info about the conditions for example temperature, photoperiod light intensity from the greenhouse conditions

Line 398: Concentration or content?

Line 416: I don’t think is necessary to mention again that was measured at 560 nm. H2O2 is content or concentration? In the text, you mentioned the "concentration" of H202 (Lines 336, 259, and 187)

Line 428-430 / 435-436: Check the sentence. You described the solution used but what did you do with that?

Line 440: The experiment was in hydroponic conditions without soil?

Line 448: Change “remained as a control” by “was kept as control” erase “,” add the before remained.

Line 451: Change "of" by “by”

Line 452: Change "of" by "from"

Line 453: Change “of” by "from"  and “was” by "were"

Line 460:  Change “deserved” by "observed" and erase “using a”

Line 461: You should mention better the instrument. It’s better … by an Axio Imager Z1 microscope Zeiss (Jena, Germany).

Line 464: Change  â—¦C by â—¦C

Line 468: Spelling (SE)

Line 479: Check the sentence. I am not sure if there are two instruments.

References:

The year from the journal has to be in bold in all references.

Line 484: Check the abbreviation of the journal

Line 501:  Probably, the editors are going to ask you for more information about the book, for example, Editorial, City and ISBN if you have it.

Line 542: Soil the first in capital letters.

Reviewer 2 Report

Dear Authors,

Reviewer comments plants-582425

The manuscript entitled „Effects of NaHCO3 acclimation on rye (Secale cereale) growth under sodic-alkaline stress“ represents a useful study aimed at an investigation of the effects of alkalinity stress (100 mM NaHCO3) on Na+/K+ ratio, Ca2+ and Mg2+ contents, oxidative stress (H2O2, MDA) and antioxidant enzymes activities in root, leaf sheath and blade in rye (Secale cereale) either non-treated (non-acclimation NA) or pre-treated with low levels of 1 and 2 mM NaHCO3. Pretreatment with low levels of 1 and 2 mM NaHCO3 significantly improved plant tolerance to enhanced NaHCO3 levels (100 mM NaHCO3).

The experimental design of the study is appropriate since it covers both control and non pretreatment (NA) vasriants as compared to pre-treated ones (A1, A2).

However, I have several comments on the manuscript:

1/ Materials and methods:

Rye (Secale cereale) cultivar (genotype, line) used in the experiments has to be given.

Information on pH value of the nutrient solution used for control treatment (no NaHCO3) has to be given in the section 4.1. on plant material and treatment conditions.

2/ Results:

In the figure and table legends, statistical test used for determination of significant differences has to be given. In Materials and methods, section Statistical analysis, the authors write that they used ANOVA analysis followed by Duncan´s multiple range test at 0.05 level; however, I think that it would be better to provide this information on the statistical test used directly in the figure legends.

In Figure 5 legend, the authors write about ten times lower NaHCO3 concentrations in NaHCO3 treatment and pre-treatments, i.e.,  10 mM NaHCO3 treatment (instead of 100 mM NaHCO3) and 0.1 and 0.2 mM NaHCO3 pre-treatments (instead of 1 and 2 mM NaHCO3), respectively, when compared to all other results. Moreover, no information on the altered levels of NaHCO3 is provided in Materials and methods. I thus recommend the authors to check the data on NaHCO3 levels in Figure 5 legend. Moreover, in Figure 5C, the authors have to add a scale bar to the photos of Evan´s Blue stained rye root tips to provide basic information on the size of the individual root tips zones.

3/ Formal comments:

Terminology: The authors use the terms „Na+ exclusion“ and „Na+ extrusion“; I recommend the authors to use the term „Na+ exclusion“ only. Please, correct „Na+ extrusion“ to „Na+ exclusion.“ (line 65 and further in the text).

Remove „the“ preceding the terms such as „Na+ concentration“ (line 279), „water uptake“ (line 310), „dry weight“ (line 395), „chlorophyll“ or „Chl“ (lines 399, 400, and further in the text, etc.), „absorbance“ (line 415), and others.

Use SI units for volume, i.e., write „350 cm3“ instead of „350 ml“ (line 379), „5 cm3“ instead of „5 ml“ (line 452), and further.

Line 449: Modify the word form „remained“ to „remaining“ in the sentence „the remaining containers“.

Introduction, line 28: Modify the second sentence as gfollows: „It is necessary to study the possibility of enhancing the crop productivity by improving the crop ability to cope with various environmental stresses.“ (NOT „It is necessary to study the possibility of enhancing the crop productivity cope with various Environmental stresses, which inhibiting the crop productivity.“)

Introduction, line 49: Add a comma following the word „Therefore,…“

Introduction, line 61: Add the word „to“ preceding the words „cope with salinity and sodic alkalinity“, i.e., „ to cope with salinity and sodic alkalinity…“

Introduction, line 65: Correct the term „Na+ exclusion“““ (NOT „Na+ extrusion“).

Introduction, line 80: Correct the plant Scientific name of „Secale cereale“ (not „Secale cereal“).

Results, line 144: Add the word „respectively“ at the end of the sentence „…and 46 and 27% lower in the leaf tissues and root tissues in the 2 mM NaHCO3 acclimated plants compared to the non-acclimated plants, respectively.“

Results, line 166: Add the word „to“ following the word „compared“ in the sentence „The strongest Evan´s Blue staining of the root tips was observed in the non-acclimated plants compared to that in the acclimated plants…“

In Figure 6B, correct the unit „H2O2 nmol g-1 FW“ at y-axis.

Discussion, line 212: Modify the word „Detailly“ to „In detail,…“

Discussion, line 243: Add a comma both preceding and following the word „finally,…“

Discussion, line 255: Modify the words „…and the product of membrane lipid eproxidation (MDA).“

Discussion, line 258: Modify the word form „The maintain“ to „The maintenance of SOD, GR, CAT, and GPOX activities…“

Discussion, line 279: Remove „the“ preceding the term „Na+ concentration“.

Discussion, line 283: Add a comma both preceding and following the word „therefore“ in the sentence „…and, therefore, the exclusion of Na+…“

Discussion, line 290: Correct the term „Na+ exclusion“ (not „Na+ extrusion“).

Discussion, line 295: Add the word „in“ in the sentence „…compared to that in non-acclimated plants…“

Discussion, line 310: Remove „the“ preceding the term „water uptake“.

Discussion, line 33, 3607: Add a comma preceding and following the word „consequently“.

Discussion, line 349, 364: Add a comma preceding and following the word „therefore“.

Discussion, line 356: Modify the word form „sdignificant“ to „significantly“ in the sentence „The significantly increased SOD activity…“

Materials and methods, line 395: Remove „the“ preceding the term „dry weight.“

Materials and methods, line 399, 400: Remove „the“ before „Chl“ for „chlorophyll“.

Materials and methods, lines 415, 464: Remove „the“ preceding the term "absorbance“.

Materials and methods, line 449: Modify the word „remained“ to „remaining“ in the sentence „…and remaining containers were treated with 10 mM NaHCO3 pH 8.05.“

Conclusions, line 473: Remove „the“ preceding the term „Na+ detoxification…“
